# Enhancing Radiologist Efficiency with AI: A Multi-Reader Multi-Case Study on Aortic Dissection Detection and Prioritization

**DOI:** 10.3390/diagnostics14232689

**Published:** 2024-11-28

**Authors:** Martina Cotena, Angela Ayobi, Colin Zuchowski, Jacqueline C. Junn, Brent D. Weinberg, Peter D. Chang, Daniel S. Chow, Jennifer E. Soun, Mar Roca-Sogorb, Yasmina Chaibi, Sarah Quenet

**Affiliations:** 1Avicenna.AI, 375 Avenue du Mistral, 13600 La Ciotat, France; angela.ayobi@avicenna.ai (A.A.); mar.roca@avicenna.ai (M.R.-S.); yasmina.chaibi@avicenna.ai (Y.C.); sarah.quenet@avicenna.ai (S.Q.); 2Department of Radiology and Imaging Sciences, Emory University School of Medicine, 1364 Clifton Road Northeast, Suite BG20, Atlanta, GA 30322, USA; colin.zuchowski@emory.edu (C.Z.); jjunn@emory.edu (J.C.J.); brent.d.weinberg@emory.edu (B.D.W.); 3Department of Radiological Sciences, University of California Irvine, Irvine, CA 92697, USA; changp6@hs.uci.edu (P.D.C.); chowd3@hs.uci.edu (D.S.C.); jesoun@hs.uci.edu (J.E.S.); 4Center for Artificial Intelligence in Diagnostic Medicine, University of California Irvine, Irvine, CA 92697, USA

**Keywords:** aortic dissection, automated detection, deep learning, prioritized worklist, emergency radiology, multi-reader multi-case study

## Abstract

Background and Objectives: Acute aortic dissection (AD) is a life-threatening condition in which early detection can significantly improve patient outcomes and survival. This study evaluates the clinical benefits of integrating a deep learning (DL)-based application for the automated detection and prioritization of AD on chest CT angiographies (CTAs) with a focus on the reduction in the scan-to-assessment time (STAT) and interpretation time (IT). Materials and Methods: This retrospective Multi-Reader Multi-Case (MRMC) study compared AD detection with and without artificial intelligence (AI) assistance. The ground truth was established by two U.S. board-certified radiologists, while three additional expert radiologists served as readers. Each reader assessed the same CTAs in two phases: assessment unaided by AI assistance (pre-AI arm) and, after a 1-month washout period, assessment aided by device outputs (post-AI arm). STAT and IT metrics were compared between the two arms. Results: This study included 285 CTAs (95 per reader, per arm) with a mean patient age of 58.5 years ±14.7 (SD), of which 52% were male and 37% had a prevalence of AD. AI assistance significantly reduced the STAT for detecting 33 true positive AD cases from 15.84 min (95% CI: 13.37–18.31 min) without AI to 5.07 min (95% CI: 4.23–5.91 min) with AI, representing a 68% reduction (*p* < 0.01). The IT also reduced significantly from 21.22 s (95% CI: 19.87–22.58 s) without AI to 14.17 s (95% CI: 13.39–14.95 s) with AI (*p* < 0.05). Conclusions: The integration of a DL-based algorithm for AD detection on chest CTAs significantly reduces both the STAT and IT. By prioritizing urgent cases, the AI-assisted approach outperforms the standard First-In, First-Out (FIFO) workflow.

## 1. Introduction

Aortic dissection (AD) is a severe thoracic aortic disorder and a cardiovascular emergency associated with a high mortality risk [1]. Its incidence is rising, increasing by 15 cases per 100,000 patients annually [2,3,4]. If left untreated, acute AD has a 33% mortality rate within the first 24 h, which increases to 50% by 48 h and can reach 75% in undiagnosed cases of ascending AD [2,5,6,7]. Nearly 22% of patients die before reaching medical care [3,4]. Timely diagnosis through imaging and prompt patient management are critical in cases of AD, as the mortality risk increases by 1–2% per hour during the first 24 h [8].

Computed tomography angiography (CTA) is considered to be the gold standard for diagnosing suspected AD due to its non-invasive nature and its ability to rapidly produce high-quality images [4]. However, the Royal College of Radiologists reports that radiologists face significant pressure due to increasing workloads and demands for efficiency, which contributes to fatigue, errors, and diagnostic delays [9,10]. Some studies reported severe consequences of treatment delays, often resulting from misdiagnosis, late diagnosis, or a low clinical index of suspicion in the emergency department [11,12].

Deep learning-based artificial intelligence (AI) systems are emerging in radiology, showing promise across several fields including neurology, cardiology, thoracic imaging, and cancer screening [13,14,15,16,17]. Our prior research on AI applications in pulmonary embolism, ASPECTS scoring, and intracranial hemorrhage has consistently demonstrated the technology’s ability to enhance diagnostic accuracy and workflow efficiency [18,19,20,21,22]. Aortic dissection—a life-threatening condition which demands rapid and accurate detection where even minor delays can substantially increase morbidity and mortality—was selected as the target disease in this study due to its critical clinical urgency and the unique potential for AI to address significant gaps in diagnostic speed and accuracy in high-stakes, time-sensitive scenarios. AI tools have been shown to aid in the identification of AD features on CTA, reducing the risk of missed lesions [23]. New AI tools have been developed and validated to assist radiologists by prioritizing cases of suspected AD, ensuring they receive urgent attention [24,25]. These AI systems effectively identify most dissections and all available aortic ruptures, placing critical cases at the top of radiologists’ worklists. This prioritization enables timely and accurate diagnosis and treatment for patients requiring immediate care by streamlining radiologist workflow [26]. This approach offers a more effective alternative to the First-In, First-Out (FIFO) methodology, which fails to account for case urgency and severity, potentially increasing patient risk [27].

There are limited studies in the literature on the advantages of computerized tools for AD detection in relation to radiologist workflow and patient outcomes. A recent study highlighted that an automated tool not only exhibited good technical performance but also significantly reduced the average time between study intake and radiologist interpretation [26]. To elucidate the clinical benefits of integrating an automated tool for AD detection and prioritization into clinical workflows, we conducted a retrospective Multi-Reader Multi-Case (MRMC) study. This study aimed to assess the impact of the validated AI algorithm CINA-CHEST for AD (Avicenna.AI, La Ciotat, France) on radiologists’ efficiency and the time required to identify AD-positive cases. We simulated two clinical workflows for AD detection: a conventional FIFO approach without AI assistance and an AI-enhanced approach based on prioritization. We hypothesized that the use of AI would play a crucial role in prioritizing critical AD cases, thereby improving the timeliness of diagnosis. 

## 2. Materials and Methods

### 2.1. Data Collection

This study was conducted in accordance with the 1975 Helsinki Declaration (as revised in 2013). Prior to investigator assessments, all data were anonymized in compliance with HIPAA and the General Data Protection Regulation (GDPR) (EU) 2016/679. Informed consent was waived when it was deemed necessary following national legislation and institutional protocols. Anonymized CTA cases were acquired between July 2017 and December 2018 from multiple clinical sites across 90 cities in the U.S. by a large U.S. teleradiology network and provided retrospectively. The dataset included CTA images from scanners manufactured by four vendors, representing 16 different scanner models: 5 from GE Medical Systems (Chicago, IL, USA), 2 from Philips (Amsterdam, The Netherlands), 6 from Siemens (Berlin, Germany), and 3 from Canon/Toshiba (Otawara, Japan). Patients were selected consecutively based on the following inclusion criteria: (1) age ≥ 18 years old, (2) chest or thoraco-abdominal CTA scans, (3) slice thickness ≤ 3 mm with no gap between successive slices, and (4) soft tissue reconstruction kernel. Exclusion criteria included the following: (1) scans that did not adhere to the recommended acquisition protocol, (2) thoracic aorta out of the field of view, and (3) significant acquisition artifacts impeding CTA interpretation.

### 2.2. The Ground Truth

Two U.S. board-certified expert radiologists (P.D.C. and D.S.C.), with 7 and 6 years of experience in clinical practice, respectively, independently reviewed the CTA images to establish the ground truth. This included determining the presence or absence of aortic dissections (ADs) and classifying them by type. All classifications of AD—hyperacute, acute, subacute, or chronic—were considered positive for AD. In case of disagreement, a third U.S. board-certified expert radiologist (J.E.S.), with 8 years of experience in clinical practice, evaluated the cases, and the ground truth was determined by majority agreement. Additionally, the radiologists documented any observed confounding factors, such as thoracic or abdominal aneurysms, intramural hematoma (IMH), calcifications, streak and motion artifacts, noise, and postoperative instances (e.g., presence of stents or grafts). 

#### AI Algorithm for AD

An FDA-approved and CE-marked commercially available DL-powered application for AD, CINA-CHEST v1.0.3 (Avicenna.AI, La Ciotat, France), was utilized in this study. This application automatically processes CTA scans and generates notifications of suspected findings, if present, alongside the corresponding image series information. For cases flagged as positive by the application, the type of AD (Type A or B) is displayed, and the dissection is identified with a red bounding box. Cases are subsequently prioritized in the radiologist’s worklist based on their positive or negative classification. A detailed explanation of the deep learning algorithm and the integration process of CINA-CHEST for AD is available in a recent study [24].

### 2.3. Multi-Reader Multi-Case (MRMC) Study

A retrospective, multi-center, fully crossed MRMC study was conducted to evaluate the clinical efficacy of CINA-CHEST for AD within clinical workflow. The study comprised two phases, namely a pre-AI phase (Unaided Arm), in which radiologists identified AD without access to the software outputs, and a post-AI phase (Aided Arm), in which radiologists identified AD assisted by the application outputs. 

Three radiologists, distinct from those who established the ground truth, participated in the study (J.C.J., B.W., and C.Z.; two U.S. board-certified radiologists; and one fellow in general radiology). These readers had 9, 5, and 2 years of experience in radiology clinical practice, respectively. The readers evaluated all CTA scans twice—once unaided and once aided by the software outputs—with a 1-month washout period between sessions to mitigate recall bias. The study design and the radiologist reading workflow are summarized in Figure 1.

In the pre-AI phase, cases were presented in random order without alerts, simulating a conventional First-In, First-Out (FIFO) worklist [27]. Radiologists evaluated the cases as they would in their routine daily practice, with each CTA appearing sequentially in the worklist and the most recently completed examination positioned at the bottom. In the post-AI phase, the AI application flagged cases suspected to be positive for AD and positioned them at the top of the worklist for evaluation (Figure 1b and Figure 2).

All three readers independently evaluated the CTAs for AD. Cases with uncertain findings were marked as indeterminate and excluded from the analysis, while cases without dissection were labeled “No dissection”. The readers were blinded to each other’s assessments, the ground truth, and patients’ clinical data.

The evaluation time for each case was automatically recorded, beginning when the reader initiated the analysis and ending when they validated their result and proceeded to the next case. Two key metrics were derived from these data. The primary endpoint, the scan-to-assessment time (STAT) for true positive cases, was defined as the cumulative duration (in minutes) from when a study became available for interpretation on the clinical workstation to the moment the final diagnosis was confirmed. The secondary endpoint was the per-case interpretation time (IT), defined as the time interval (in minutes) from when a radiologist opened the corresponding CTA to when they submitted their final diagnosis. This metric represents the time required for a radiologist to analyze, interpret, and confirm their findings for a single case before proceeding to the next.

The effectiveness of CINA-CHEST for AD in reducing the time required to identify and assess AD on CTA was evaluated under the assumption that all the scans were previously acquired and simultaneously became available for interpretation on the workstation. This simulates a high-workload emergency department, representing a situation where rapid AD identification is essential.

### 2.4. Statistical Analysis

Initially, the results computed by CINA-CHEST for AD were compared to the ground truth, with the area under the receiver operating characteristic curve (AUROC) as well as sensitivity, specificity, and accuracy being calculated for the entire dataset. The 95% confidence intervals (CIs) for sensitivity, specificity, and accuracy were determined using the Clopper–Pearson method based on the exact binomial distribution. The performance of each of the three U.S. board-certified radiologists was first evaluated individually against the ground truth and then assessed across both phases of the study—pre- and post-AI implementation—by measuring the AUROC, sensitivity, specificity, and accuracy in each phase. 

To compare workflow metrics pre- and post-AI, the STAT for each true positive case was calculated for each arm as follows (Equation (1)):(1)$$\mathrm{STAT}(n)\;\mathrm{in}\;\mathrm{minutes}=\sum^{n}i=1\mathrm{IT}(i),$$
where STAT(*n*) represents the scan-to-assessment time for the *n*-th case and IT(*i*) denotes the interpretation time for the *i*-th case. Stratified analyses were conducted to explore potential variations based on readers’ experience, with a per-group comparison of junior (<5 years of clinical experience) versus senior readers (≥5 years of experience). Additionally, a comparison of STAT between pre- and post-AI phases including all positive and negative cases was performed. Per-case IT was calculated for each arm as follows (Equation (2)):

IT (seconds) = T*_Diagnosis_* − T*_study open_*.
(2)


The differences between the aided and unaided arms for mean STAT and mean IT were assessed.

To evaluate the statistically significant difference (significant reduction, α ≤ 0.05, two-sided) between the aided and the unaided arms, a mixed-effects repeated measures model was implemented. Reader, case, and AI usage (aided vs. unaided) were included as fixed-effect terms in the model and a paired-sample *t*-test was conducted [28]. A *p*-value < 0.05 was considered to represent statistical significance. All statistical analyses were conducted using MedCalc Statistical Software (v22.023, MedCalc Software Ltd., Ostend, Belgium).

## 3. Results

### 3.1. AI’s and Readers’ Performance

Each reader reviewed a total of 100 CTAs (65 negatives and 35 positives according to the ground truth) in the pre-AI phase and post-AI phase. Five negative cases were marked as indeterminate and thus excluded from the final cohort, resulting in a final analysis of 285 CTA cases (95 cases per reader, including 35 positive cases) for each arm. Table 1 summarizes the distribution of cases by scanner manufacturer. The mean patient age was 58 years ± SD = 14.7, and 52.63% of the patients were male. Among the included cases, 36% presented confounding factors such as intramural hematoma (IMH), aortic wall calcifications, aneurysms, and the presence of stents, grafts, or streak/motion artifacts. Additionally, pulmonary embolism was reported in 4% of cases. This study also considered different dissection types and complexities, with 15% being Type B dissection (which occurs in the descending aorta), which is subtle and challenging, and 20% being Type A dissection (which originates in the ascending aorta), which is larger and generally easier to detect. The software misclassified CTAs in 2 out of 95 patients (2.1%), resulting in 2 false negatives and 33 true positives. The first false negative result was attributed to the presence of both IMH and a graft or stent, while the second was due to the presence of IMH alone. When compared to the ground truth, CINA-CHEST for AD demonstrated an AUROC of 0.971 (95% CI: 0.915–0.995), an accuracy of 97.89% (95% CI: 92.6–99.74%), a sensitivity of 94.29% (95% CI: 80.84–99.3%), and a specificity of 100% (95% CI: 94.04–100.00%).

The accuracies of the three readers during the aided phase (using CINA-CHEST for AD) were 97.89% (95% CI: 92.6–99.74%), 97.89% (95% CI: 92.6–99.74%), and 98.94% (95% CI: 94.27–99.97%), respectively. In comparison, their accuracies during the unaided phase (without the software) were 98.95% (95% CI: 94.33 to 99.97%), 97.89% (95% CI: 94.33 to 99.74%), and 97.89% (95% CI: 92.60–99.74%), respectively. The sensitivity, specificity, and AUROC of each reader across each phase of the study are shown in Table 2. The statistical analysis found no significant difference in performance between the two phases (*p* > 0.05). All readers misclassified one case as positive in the pre-AI phase, which was correctly identified as a true negative during the post-AI phase. Additionally, two of the three readers classified two cases as indeterminate in the pre-AI phase; these cases were later identified as true negatives in the post-AI phase but were excluded from the final analysis.

### 3.2. Comparison of Scan-to-Assessment Times

#### 3.2.1. STAT for AD True Positives Cases

An analysis was conducted to compare the STATs for true positive AD cases before and after the implementation of the AI software v1.0.3. This analysis evaluated the efficiency of time to assessment across 33 true positive AD cases per reader per arm, resulting in a total of 99 cases analyzed in both the pre- and post-AI phases.

The results of the analysis across the three readers demonstrated a significant reduction in the STAT with AI assistance compared to the unaided arm (*p* < 0.01). In the pre-AI phase, the mean STAT for true positive cases was 15.84 min (SD: 12.38 min; 95% CI: 13.37–18.31 min). In the post-AI phase, the mean STAT was notably reduced to 5.07 min (SD: 4.24 min; 95% CI: 4.23–5.91 min). This represents a decrease of 10.77 min (SD: 12.96 min; 95% CI: −13.36 to −8.18 min) (Figure 3).

Furthermore, a per-reader analysis of true positive AD cases indicated that each reader demonstrated a significant reduction in the mean STAT between the pre- and post-AI phases (*p* < 0.05). The magnitude of reduction varied across readers, ranging from 5.08 (SD: 7.05 min; 95% CI: −7.64 to −2.53 min) to 15.19 min (SD: 14.69 min; 95% CI: −20.40 to −9.98 min) (Figure 4 and Table 3).

Finally, a per-group analysis on true positive AD cases comparing junior versus senior readers revealed a significant reduction in the STAT for both groups (*p* < 0.05). For the junior group, the difference between pre- and post-AI conditions was −5.08 min (SD: 7.05 min; 95% CI: −7.64 to −2.53 min). For the senior group, the difference was −13.63 min (SD: 14.25 min; 95% CI: −17.12 to −10.13 min) (Table 4).

#### 3.2.2. STAT for All Cases

The STAT was evaluated across all 95 cases, including both AD positives and negatives, to assess the overall impact of AI implementation. The results reflect the combined performance of all three readers. In the pre-AI phase, the mean STAT was 17.17 min (SD: 12.16 min; 95% CI: 15.75–18.60 min). With AI software assistance, the mean STAT was reduced to 12.54 min (SD: 7.15 min; 95% CI: 11.71–13.86 min). This reduction of approximately 4.62 min (SD: 13.06 min; 95% CI: −6.14 to −3.10 min) was statistically significant (*p* < 0.01) (Table 5).

A per-reader STAT analysis was also performed for all cases. The first reader showed a difference of 0.45 min or 27.26 s (SD: 7.78 min; 95% CI: 1.13–2.04 min) in the mean STAT between the pre- and post-AI conditions, which was not statistically significant (*p* > 0.05). In contrast, the second and third readers exhibited statistically significant reductions in the STAT between the two study phases: −9.84 min (SD: 14.33 min; 95% CI: −12.76 to −6.92 min) (*p* < 0.01) and −4.48 min (SD: 13.33 min; 95% CI: −7.33 to −1.63 min) (*p* < 0.01), respectively (Table 5).

### 3.3. Comparison of Per-Case Interpretation Time (IT)

The IT was evaluated for a total of 570 cases (95 cases per arm, per reader). The mean per-case IT was 21.22 s or 0.35 min (SD: 11.62 s; 95% CI: 19.87–22.58 s) in the pre-AI phase, whereas in the post-AI (Aided) phase, the value decreased to 14.17 s or 0.24 min (SD: 6.7 s; 95% CI: 13.39–14.95 s). This reduction in IT represents a statistically significant difference of 7.04 s or 0.11 min (*p* < 0.01) (Figure 5).

## 4. Discussion

In this retrospective multi-center, fully crossed MRMC study, three radiologists independently interpreted 95 cases twice—once aided and once without AI assistance—to assess the clinical efficacy of CINA-CHEST for AD. The primary objective was to determine whether the use of an AI tool could reduce the STAT for AD-positive cases compared to the traditional FIFO workflow. The findings highlight the challenges radiologists face in promptly identifying urgent cases within high-pressure, high-volume clinical environments. To the best of our knowledge, this study is the first to demonstrate improved STAT and IT in the identification and prioritization of AD on CTA scans with the integration of an AI device.

The CINA-CHEST for AD software demonstrated strong standalone performance, achieving an AUROC of 0.971, an accuracy of 97.89%, a sensitivity of 94.29%, and a specificity of 100%. Radiologist performance remained consistently high with the use of the software. The junior reader maintained 100% sensitivity across both phases, with only a minor, non-significant drop in specificity (98.33% to 96.66%). Both senior readers showed slight improvements in accuracy in the post-AI phase, with one of them attaining 98.94% accuracy compared to 97.89% pre-AI. Across all readers, the overall AUROC remained consistently high in both phases, highlighting the software’s ability to support accurate diagnosis without significantly altering radiologist performance. Importantly, in the pre-AI phase, one case was misclassified as positive by all readers, which was correctly identified as a true negative with the aid of the device in the post-AI phase. Furthermore, two of the three readers classified two cases as indeterminate in the pre-AI phase; these cases were later confirmed as true negatives in the post-AI phase but were excluded from the final analysis. These findings emphasize the diagnostic accuracy and reliability of the AI tool, demonstrating its potential to enhance radiologist workflow without introducing variability in clinical judgment.

By simulating real-world clinical workflow, this study demonstrates that AI expedites the identification and prioritization of critical AD cases. With AI-assisted detection, radiologists identified all positive cases in an average of 5.07 min, approximately 11 min faster than with the traditional FIFO workflow, resulting in a 68% improvement in efficiency. The non-overlapping 95% confidence intervals between the pre- and post-AI phases provide compelling evidence that this time reduction is not due to random variations or outliers, but rather a robust effect of AI support, reinforcing the clinical relevance of AI integration in time-sensitive and busy diagnostic workflows. Furthermore, by incorporating a balanced mix of subtle and complex cases, the study evaluated the AI tool’s ability to detect both straightforward and challenging presentations. This approach underscores the potential of AI to assist in identifying subtle findings that might otherwise be overlooked, addressing a key limitation in traditional diagnostics.

Moreover, a global comparison of STAT, encompassing all positive and negative cases, resulted in a significant reduction of 26.8% (4.6 min) in the aided arm, demonstrating that AI not only streamlines the detection of critical cases but also enhances overall workflow efficiency for both urgent and routine cases.

Prioritizing radiology worklists has the potential to enhance patient care and alleviate radiologist workload in contrast to the traditional FIFO workflow, which often relies upon incomplete and ambiguous priority categories (e.g., stat, ASAP, now, and critical) defined by the ordering physician’s urgency assessment [29]. By actively prioritizing cases flagged as positive, AI may enable radiologists to allocate more time to critical instances, which are frequently misdiagnosed due to their nonspecific presentation [25,26]. According to the International Registry of Acute Aortic Dissection, the median time from emergency department presentation to a definitive diagnosis of acute aortic dissection is 4.3 h, largely attributed to high patient volumes [30,31]. The implementation of an AI algorithm capable of detecting AD features on CT images could significantly reduce delays in diagnosing and treating serious aortic lesions and shorten hospital stays for patients [17,32].

To date, no other studies have specifically addressed the impact of the prioritization of AD detection on the STAT, although the rapid progression and severe outcomes associated with AD make it an ideal candidate to showcase AI’s potential for real-time decision support in high-stakes scenarios. Previous research has primarily assessed the diagnostic performance of AI triage solutions for AD detection, often comparing the algorithm outputs to those of radiologists. However, such studies provide limited insight into the broader clinical benefits of AI on radiology workflow efficiency and patient management [24,33,34,35]. Harris et al. developed a convolutional neural network model trained to detect AD and rupture, resulting in a median reduction of 395 s in the delay time—the interval between when a study is received by the system and when it is opened by a radiologist. While this study begins to explore the role of prioritization in AD detection, its primary focus is on evaluating the technical processing performances of the algorithm [26].

Conversely, several studies have evaluated the effectiveness of AI-based prioritization for alternative pathologies, such as pulmonary embolism (PE), intracranial hemorrhage (ICH), and cancer. For example, AI-prioritized worklists have been shown to significantly reduce the time to diagnosis of incidental PE on CT scans in patients with cancer. The median turnaround time (TAT) for true positive examinations flagged by the AI software was reduced to 91 min—a significant improvement from several days to 1.5 h compared to the traditional FIFO workflow [36]. Moreover, the implementation of AI in the radiology workflow reduced the scan-to-alert time (from scan initiation to AI alert) for PE cases, with an average AI alert time of under 6 min. These findings highlight the critical importance of prioritization models, which potentially improve patients’ chances of survival [18]. Similarly, for ICH detection, the incorporation of a machine learning algorithm into the clinical radiology workflow significantly decreased the mean report TAT from 75 to 69 min in emergency settings, expediting critical case identification and improving patient outcomes in urgent care scenarios [37]. While these studies focus on pathologies other than AD, they demonstrate that AI-based prioritization can significantly reduce TATs and improve patient outcomes. These findings highlight the broader potential for similar benefits in various diagnostic areas.

A stratified analysis of individual reader performance revealed variability in the impact of AI integration. In an analysis including both positive and negative cases, one of the three readers demonstrated no significant difference between the aided and unaided arms and even a negligible, non-significant increase in the STAT. However, despite these individual discrepancies, the overall STAT was statistically reduced when averaged across all readers. This variability reflects differences in how AI integration may affect individual workflows. An analysis of exclusively true positive cases revealed that junior and senior readers alike demonstrated a significant reduction in the STAT during the post-AI phase, with reductions ranging from 5 to 15 min compared to the pre-AI phase. Notably, the senior readers experienced a greater STAT improvement (71.5%) than the junior group (53.8%), suggesting that more experienced radiologists derived greater efficiency gains from AI implementation compared to their less experienced counterparts. This observation aligns with findings from previous studies evaluating the IT. Bennani et al. reported greater IT improvements for general radiologists (34%) compared to residents (30%) in an MRMC study on thoracic abnormalities. Muller et al. observed a small increase in IT for one resident with AI aid; however, residents reported a better overview of cases [38]. The greater efficiency gains observed among senior radiologists suggests that their advanced skills and familiarity with complex cases enable them to leverage AI tools more effectively than junior radiologists. These results highlight that the impact of AI integration is shaped by individual user experience and clinical context.

Finally, a focused analysis was conducted on the IT, defined as the average time radiologists spend interpreting a case regardless of its position within the worklist. Our findings indicate that the aid of the AI software reduced the IT by 33%, underscoring its utility in facilitating AD detection and the identification of complex cases, enabling radiologists to dedicate more time to critical cases. These results are consistent with prior paired design studies, which reported reductions in the IT for chest CT scans of 7–44% when utilizing AI to detect and measure lung nodules [39]. Similarly, a prospective study of 390 patients observed a 22.1% reduction in the IT when cardiothoracic radiologists interpreted chest CT exams with AI assistance. In this study, the AI system automatically labeled, segmented, and measured both normal structures and abnormalities across the cardiac, pulmonary, and musculoskeletal systems [40]. Collectively, these findings suggest that AI automation enhances radiological efficiency by streamlining diagnostic processes and optimizing workflows.

In summary, the significant reductions in both the IT and STAT demonstrate the transformative impact of AI on radiology. By streamlining diagnostic processes and improving the STAT, AI enhances efficiency while shifting the paradigm from traditional FIFO methods to more effective prioritization strategies. This innovation highlights AI’s remarkable potential to revolutionize radiology workflows, solidifying its role as an invaluable tool in modern medical practice. Conducted across multiple clinical sites, scanner manufacturers, and countries, this reader study benefits from a diverse dataset encompassing a wide range imaging parameters and patient profiles. Moreover, it engaged a panel of readers representing different expertise levels, reflecting the real-world diversity of clinical practice.

This study has a few limitations. First, the workflow impact of the AI tool was assessed using a small cohort of three radiologists with varying levels of experience, which may not fully capture the variability in performance among a broader group of radiologists. Future research should evaluate AI’s impact across a larger group of radiologists to enhance generalizability. Additionally, the dataset analyzed comprised only 100 cases, with 5% being excluded due to indeterminate responses. Expanding the sample size in future studies would improve the statistical power and reliability of the findings. Finally, conducting a prospective study would provide a more robust validation of these tools in a real-world setting.

## 5. Conclusions

In conclusion, our MRMC study is the first to demonstrate the positive impact of AI on clinical workflow for detecting AD. The AI tool significantly reduced the time for radiologists to identify positive cases in the emergency department, enabling the prioritization of these critical cases. Given the importance of timely diagnosis and intervention in improving outcomes for patients with AD, this technology offers a transformative solution by automatically flagging and prioritizing suspected cases. By enhancing workflow efficiency, the AI tool enables radiologists to focus on the most urgent cases first, ultimately contributing to improved patient care in emergency medical settings. 

## Figures and Tables

**Figure 1 diagnostics-14-02689-f001:**
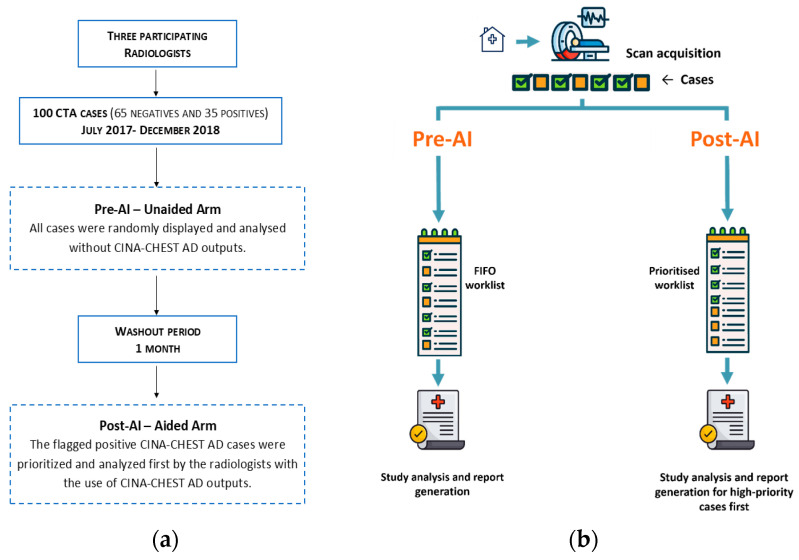
(**a**) The current study design overview. (**b**) A workflow diagram illustrating the traditional radiologist reading process alongside the AI-assisted approach.

**Figure 2 diagnostics-14-02689-f002:**
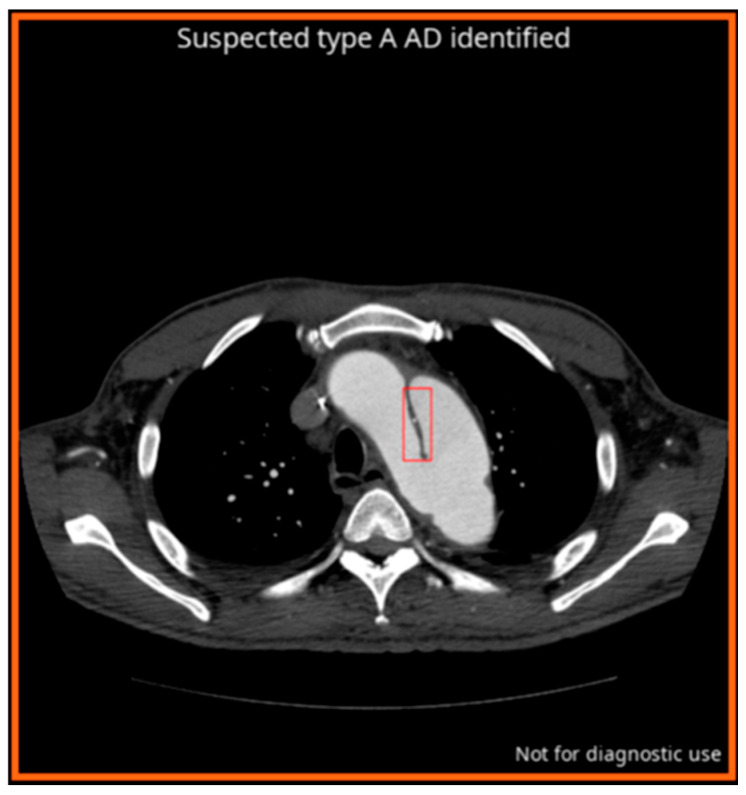
CINA-CHEST for aortic dissection (AD) outputs. The red bounding box shows the localization of the automatically detected AD. The type of AD detected is mentioned as follows: “Suspected type A AD identified”.

**Figure 3 diagnostics-14-02689-f003:**
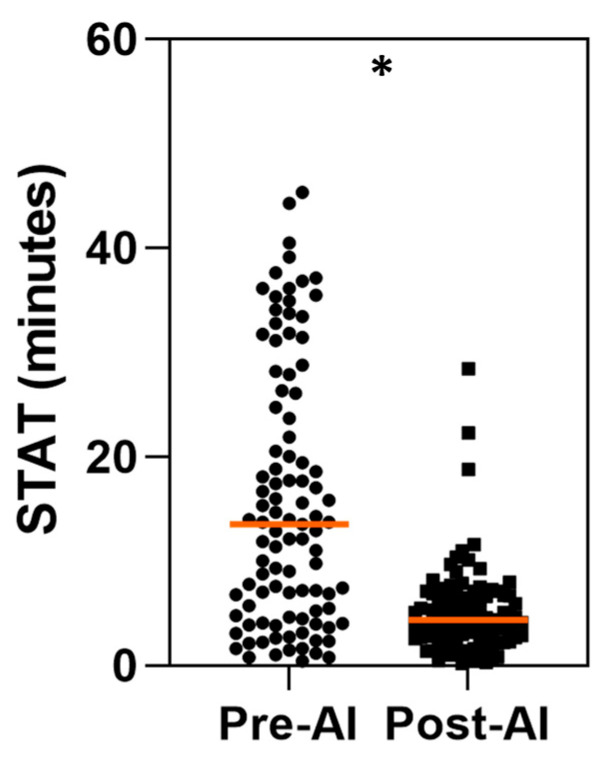
Comparison of scan-to-assessment time (STAT) only for AD true positive cases before and after AI implementation. STAT (in minutes) was measured for true positive AD cases in both pre- and post-AI phases based on assessments by three independent readers evaluating 33 CTA scans per arm. Each data point represents STAT for individual case, with central line indicating median. * *p* < 0.05 for statistically significant difference between two conditions according to paired *t*-test.

**Figure 4 diagnostics-14-02689-f004:**
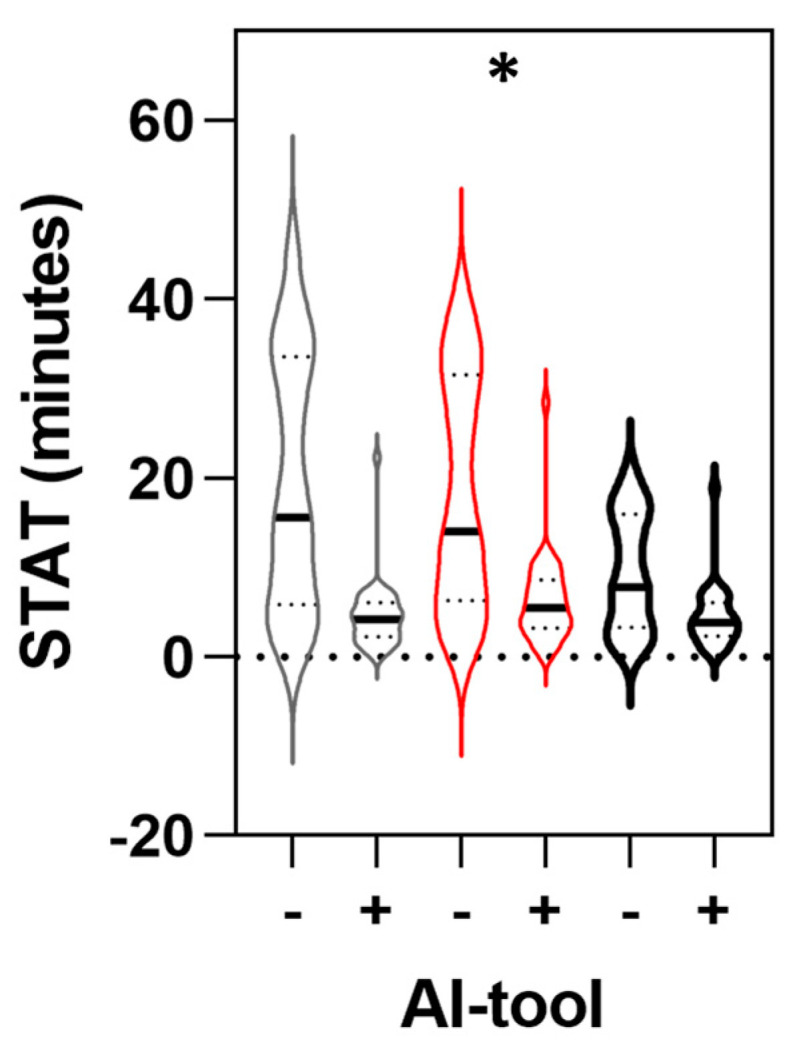
Per−reader comparison of scan-to-assessment time (STAT) for true positive AD cases before and after AI implementation. STAT (in minutes) was measured for true positive AD cases in pre-AI (−) and post-AI (+) phases. Three independent readers (represented by a different color in the figure) evaluated 33 CTA scans per condition. Main central line corresponds to median value. * *p* < 0.05 for statistically significant difference between two conditions according to paired *t*-test.

**Figure 5 diagnostics-14-02689-f005:**
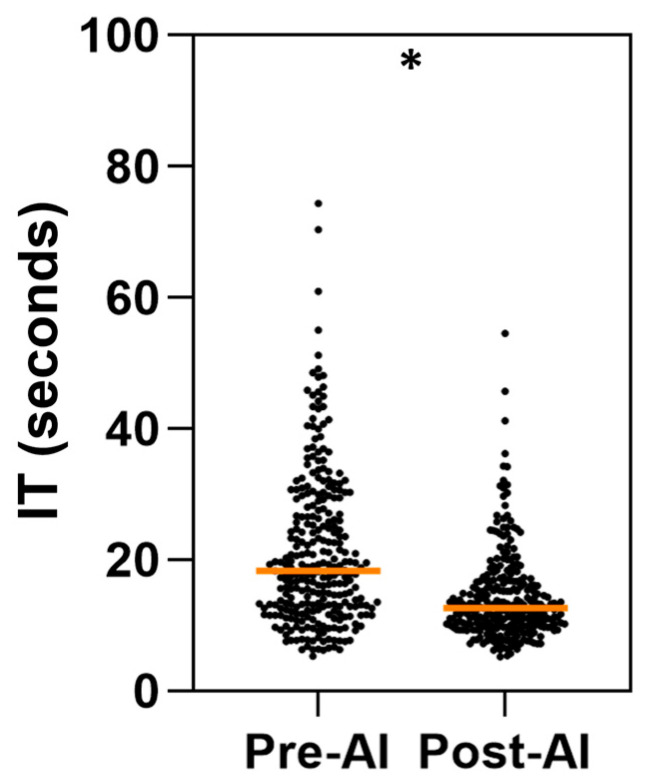
Per-case interpretation time (IT) in pre-AI and post-AI phases. IT times (in seconds) measured for both phases based on assessments from three independent readers across 95 CTA scans per condition. Central line corresponds to median value. * *p* < 0.05 for statistically significant difference between two conditions according to paired *t*-test.

**Table 1 diagnostics-14-02689-t001:** Data characteristics. Scanner makers and slice thickness distributions.

Scanner Makers	Occurrence (%)	Slice Thickness	Occurrence (%)
GE MEDICAL SYSTEMS	59 (62.11%)	ST < 1 mm	4 (4%)
SIEMENS	21 (22.1%)	1 ≤ ST ≤ 2.5 mm	83 (87%)
CANON (Formerly TOSHIBA)	10 (10.53%)	ST ≤ 3 mm	8 (9%)
PHILIPS	5 (5.26%)		
**Total 95**			

**Table 2 diagnostics-14-02689-t002:** The readers’ performances without and with CINA-CHEST for AD identification. The results include 95 CTA readings per reader and per arm. AUROC: area under the receiver operating characteristics curve.

Parameter % [95% CI]	Reader 1		Reader 2		Reader 3	
	Pre-AI	Post-AI	Pre-AI	Post-AI	Pre-AI	Post-AI
**Accuracy**	98.95% [94.33–99.97%]	97.89% [92.60–99.74%]	97.895% [92.60–99.74%]	97.895% [92.60–99.74%]	97.895% [92.60–99.74%]	98.94% [94.27–99.97%]
**Sensitivity**	100% [89.99–100.0%]	100% [89.99–100.0%]	100% [89.99–100.0%]	94.27% [80.84–99.3%]	97.143% [85.08–99.92%]	97.143% [85.08–99.92%]
**Specificity**	98.33% [91.20–99.96%]	96.66% [88.47–99.59%]	96.66% [88.47–99.59%]	100% [94.03–100.0%]	98.33% [91.20–99.96%]	100% [94.03–100.0%]
**AUROC**	0.992 [0.947–1.0]	0.983 [0.933–0.999]	0.983 [0.933–0.999]	0.971 [0.915–0.995]	0.977 [0.924–0.997]	0.986 [0.937–0.999]

**Table 3 diagnostics-14-02689-t003:** Comparison of scan-to-assessment time (STAT) for true positive AD cases between unaided and aided arms. Overall analysis across all readers (*n* = 3 radiologists) includes total of 99 cases for each arm and per-reader comparison with 33 cases per arm. Results are expressed in minutes. * Difference is statistically significant (*p* < 0.05) according to paired *t*-test.

STAT for True Positive AD Cases	Unaided Arm Time (min) Mean ± SD [95% CI]	Aided Arm Time (min) Mean ± SD [95% CI]	Aided–Unaided Difference (min) Mean ± SD [95% CI]
**All readers** (*n* = 99)	15.84 ± 12.38 [13.37, 18.31]	5.07 ± 4.24 [4.23, 5.91]	−**10.77** * ± 12.96 [−13.36, −8.18]
**Reader 1** (*n* = 33)	9.45 ± 6.41 [7.17, 11.72]	4.36 ± 3.36 [3.17, 5.56]	−**5.08** * ± 7.05 [−7.64, −2.53]
**Reader 2** (*n* = 33)	19.72 ± 14.09 [14.06, 25.38]	4.53 ± 3.83 [3.17, 5.88]	−**15.19** * ± 14.69 [−20.40, −9.98]
**Reader 3** (*n* = 33)	18.36 ± 12.29 [13.79, 22.94]	6.32 ± 5.06 [4.53, 8.10]	−**12.04** * ± 13.85 [−18.62, −5.46]

**Table 4 diagnostics-14-02689-t004:** Per-group comparison of scan-to-assessment time (STAT) in junior group versus senior group for the aided and unaided arm. All readers were taken into account (*n* = 3 readers) with total of 33 true positive cases per arm. Results are expressed in minutes. * Difference is statistically significant (*p* < 0.05) according to paired *t*-test for mean difference.

Readers’ Experience	Unaided Arm Time (min) Mean ± SD [95% CI]	Aided Arm Time (min) Mean ± SD [95% CI]	Aided–Unaided Difference (min) Mean ± SD [95% CI]
**Junior** (*n* = 33)	9.45 ± 6.41 [7.17, 11.72]	4.36 ± 3.36 [3.17, 5.56]	−**5.08** * ± 7.05 [−7.64, −2.53]
**Senior** (*n* = 66)	19.04 ± 13.41 [15.74, 22.67]	5.43 ± 4.54 [4.31, 6.54]	−**13.63** * ± 14.25 [−17.12, −10.13]

**Table 5 diagnostics-14-02689-t005:** A comparison of the scan-to-assessment time (STAT) for all cases between the aided and unaided arms. The overall analysis across all readers (*n* = 3 radiologists) includes a total of 285 cases for each arm. A per-reader comparison was conducted with 95 cases (positives and negatives) per arm. The results are expressed in minutes. * The difference is statistically significant (*p* < 0.05) according to the paired *t*-test.

STAT for All Cases	Unaided Arm Time (min) Mean ± SD [95% CI]	Aided Arm Time (min) Mean ± SD [95% CI]	Aided–Unaided Difference (min) Mean ± SD [95% CI]
**All readers** (*n* = 285)	17.17 ± 12.16 [15.75, 18.60]	12.54 ± 7.15 [11.71, 13.86]	−**4.62** * ± 13.06 [−6.14, −3.10]
**Reader 1** (*n* = 95)	10.18 ± 6.10 [8.94, 11.43]	10.64 ± 5.55 [9.51, 11.76]	0.45 ± 7.78 [1.13, 2.04]
**Reader 2** (*n* = 95)	21.46 ± 13.50 [18.71, 24.21]	11.62 ± 6.61 [10.28, 13.04]	−**9.84** * ± 14.33 [−12.76, −6.92]
**Reader 3** (*n* = 95)	19.85 ± 12.32 [17.34, 22.36]	15.37 ± 8.22 [13.70, 17.94]	−**4.48** * ± 13.33 [−7.33, −1.63]

## Data Availability

The study data are the property of Avicenna.AI and are not publicly accessible. They can be obtained from the corresponding author upon reasonable request and with the approval of the Regulatory Affairs Department of Avicenna.AI.

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
