# Peer review of "Enhancing Radiologist Efficiency with AI: A Multi-Reader Multi-Case Study on Aortic Dissection Detection and Prioritization"

_diagnostics, 2024, doi:10.3390/diagnostics14232689_

Round 1
Reviewer 1 Report
Comments and Suggestions for Authors
This is a study about the use of AI helps the diagnosis of aortic dissection (AD) cases from the images of computed tomography (CT). The authors demonstrated integrating a deep learning-based algorithm for AD detection on chest CT significantly reduces both scan-to-assessment times (STAT) and interpretation times (IT). The results finely demonstrated the benefit of AI-supported reading system of CT in AD. There were several issues to be addressed.
# It seems possible that the time required for CT reading can be shortened even without the support of AI if it is the second reading. The reverse order should also be considered.
# What was the reason that AD was selected for target disease of AI-supported CT reading? What is other disease candidate for presenting the usefulness of AI-supported CT reading?
Comments on the Quality of English LanguageNo comment
Author Response
- # It seems possible that the time required for CT reading can be shortened even without the support of AI if it is the second reading. The reverse order should also be considered.
Dear Reviewer, thank you for your insightful comments.
Regarding the time required for CT reading, we observed a significant reduction in Scan-to-Assessment Time (STAT) for detecting true positive AD cases with AI support, from a mean of 15.84 minutes (95% CI: 13.37–18.31 minutes) without AI assistance to 5.07 minutes (95% CI: 4.23–5.91 minutes) with AI assistance, as presented in section 3.2.1, lines 243-248. The non-overlapping 95% confidence intervals between these two phases provide compelling evidence that this time reduction is not due to random variation or outliers, but rather a robust effect of AI support, reinforcing the clinical relevance of AI integration in time-sensitive and busy diagnostic workflows. This information has been incorporated into the manuscript in Section 4 - Discussion, lines 345-348, and is highlighted in yellow for clarity.
Furthermore, when we consider the Interpretation Time (IT), it also decreased significantly, from 21.22 seconds (95% CI: 19.87–22.58s) without AI to 14.17 seconds (95% CI: 13.39–14.95s) with AI (p < 0.05), as shown in section 3.3, lines 305-309. This analysis does not consider the order of lecture of cases, it takes into account the time spent by the radiologist to interpret each case within the two phases of the study. Hence, a significant reduction in the IT when using AI indicates that, even without prioritising cases at the top of the list, AI outputs provide a significant reduction on the time a clinician spends analysing a case.
Following your suggestions, we recomputed the statistics under the assumption that the AI's prioritisation was not applied. In phase 2, we reorganised the cases, assuming they had been presented in a random order, consistent with phase 1. If we analyse both positive and negative cases with reverse order, a significant reduction in STAT for detecting all cases with AI assistance is observed (p<0.01). The mean time, between the three readers, decreased from 17.34 minutes (95% CI: 15.90–18.79 minutes) without AI to 11.44 minutes (95% CI: 10.60–12.27 minutes) with AI assistance. The difference was significant also for all three readers individually. For the first reader, the STAT reduction between the pre-AI and post-AI phases was -1.16 minutes (SD: 0.79 min; 95% CI: -1.32 to -0.99 min) which was statistically significant (p < 0.01). The second and third readers also demonstrated significant differences between the two study phases: 10.54 minutes (SD: 7.27 min; 95% CI: -12.02 to -9.06 min) and 6.0 minutes (SD: 4.1 min; 95% CI: -6.84, - 5.17 min), respectively (p < 0.01). These results are consistent with results described in the article section 3.2.2 STAT for all cases and table 5.
In addition, when analysing the random order in the second reading (without prioritisation), which included only 33 true positive cases per reader for a total of 99 cases, the results demonstrated a significant reduction in STAT with the use of the software compared to the unaided arm (p < 0.01). In the pre-AI phase, the mean STAT for true positive cases was 15.84 minutes (SD: 12.38; 95% CI: 13.37–18.31). Following the implementation of AI, the mean STAT decreased significantly to 10.55 minutes (SD: 7.31; 95% CI: 9.09–12.01), representing a reduction of 5.3 minutes (SD: 5.9; 95% CI: -6.46 to -4.10).
Although these results differ from those in Section 3.2.1, where STAT was calculated for AD true positive cases based on AI prioritisation, showing a reduction of 10.77 minutes, the observed decrease between pre- and post-AI phases in this analysis remains significant. This improvement is particularly critical for addressing life-threatening pathologies.
We would like to clarify that the results from the analysis without prioritization, while insightful, were not integrated into the main manuscript because the primary focus of our study is on the impact of AI prioritisation on the detection of true positive cases. The scope of the study was specifically designed to assess how AI-driven prioritization affects diagnostic workflows and time efficiency for life-threatening conditions, particularly positive cases of aortic dissection.
Including the additional analysis on randomised case order could risk shifting the focus away from this primary objective. However, if you believe it would be beneficial for the clarity or comprehensiveness of the manuscript, we would be happy to incorporate this analysis as supplementary material or discuss it in a dedicated section. We value your input and are eager to ensure that our work aligns with your recommendations. Please let us know if this approach is acceptable to you.
2. # What was the reason that AD was selected for target disease of AI-supported CT reading? What is other disease candidate for presenting the usefulness of AI-supported CT reading?
Dear Reviewer,
Thank you for your interesting comments. We appreciate the opportunity to clarify our rationale for selecting Aortic Dissection (AD) as the focus of this study.
AD was chosen due to its critical clinical urgency and the potential for AI to address a significant diagnostic challenge. As a life-threatening condition, AD demands rapid and accurate detection; even minor delays can substantially increase morbidity and mortality. The complexity and time-sensitive nature of AD diagnosis highlight the need for innovative tools like AI to enhance the speed and reliability of imaging evaluations, thereby improving patient outcomes.
Our team has previously explored AI applications in conditions such as Pulmonary Embolism, ASPECTS scoring, and Intracranial Haemorrhage. These studies have consistently demonstrated AI's efficacy in improving diagnostic accuracy and workflow efficiency, with findings validated in both clinical practice and scientific literature (1–5, among others). Additionally, AI is achieving maturity in detecting other pathologies, such as lung nodules, breast and prostate cancer, with well-validated tools successfully integrated into clinical workflows.
In contrast, AD represents a domain of unmet need, where current diagnostic approaches struggle to deliver the speed and accuracy required for optimal patient care. The rapid progression and severe outcomes associated with AD make it an ideal candidate to showcase AI's potential for real-time decision support in high-stakes scenarios. Our aim was to demonstrate AI's ability to reduce diagnostic time and improve efficiency for radiologists managing AD in busy clinical settings.
Some of the requested information was already addressed in the manuscript (lines 54–56). To provide further context, additional details have been incorporated into Section 1 – Introduction (lines 56–63), Discussion (lines 370-372) and highlighted in yellow for clarity.
3. #The English could be improved to more clearly express the research.
To ensure the highest quality of language and clarity in our manuscript, we had it thoroughly reviewed and refined by a native English speaker. We hope this enhances the readability and precision of our work.
Please see the attachment

Reviewer 2 Report
Comments and Suggestions for Authors
In this paper, the authors assessed the ability of AI tool to diagnose aortic dissection, a cardiac emergency that needs prompt diagnosis to avoid grave consequences.
The authors found that the AI was very robust in the diagnosis of AD compared to expert readers. Despite the small number of cases, the study is well designed and conducted. The paper I’d well written and easy to read and the authors have discussed the potential limitations of the study. However, it would be interesting to know the grade of difficulty of these case and how extensive the dissection is. This can be graded by the expert reader. We know that small dissection may be messed or require further diagnostic tests.
Author Response
- # It would be interesting to know the grade of difficulty of these cases and how extensive the dissection is. This can be graded by the expert reader.
Dear Reviewer, thank you for your insightful comments.
In this study, 36% of the cases presented confounding factors such as intramural hematoma (IMH), aortic wall calcifications, aneurysms, and the presence of stents, grafts, or streak/motion artifacts. Additionally, pulmonary embolism was reported in 4% of cases. The inclusion of these challenging scenarios was deliberate, aiming to evaluate the AI tool’s capability to address and overcome complexities encountered in real-world clinical settings.
This information was provided by three expert readers who established the (GT), who detailed the challenges they faced during classification, including the detection of pulmonary embolism. The Section 2.2: The Ground Truth explain that the GT documented any observed confounding factors, such as thoracic or abdominal aneurysms, intramural hematoma (IMH), calcifications, streak and motion artefacts, noise, and postoperative instances (e.g., presence of stents or grafts).
Additional information has been incorporated into Section 3.1, line 207-210 and highlighted in yellow in the manuscript.
Regarding how extensive the dissections were, please, refer to the answer 2.
2. We know that small dissection may be messed or require further diagnostic tests.
Dear Reviewer,
Thank you for your observation. We agree that smaller dissections can be more easily missed or may require additional diagnostic tests for confirmation.
In this study, our focus was on the type and complexity of the cases rather than dissection size. Specifically, 15% of cases were Type B dissections, which occur in the descending aorta, a structurally confined segment often presenting unique diagnostic challenges due to their subtle and variable nature. Meanwhile, 20% of cases were Type A dissections, which originating in the ascending aorta, are typically larger, often involving the aortic arch and descending aorta, and are generally easier to detect due to their size and location.
By including a balanced mix of subtle and complex cases, we aimed to evaluate the AI tool’s capability to detect both obvious and challenging presentations. This approach underscores the potential of AI to assist in identifying subtle findings that might otherwise be overlooked, addressing a key limitation in traditional diagnostics.
This information has been incorporated into Section 3.1, line 210-213 and highlighted in yellow in the manuscript and in Section 4 Discussion, lines 348-352.
Please see attached
